# Hesperidin Reverses Oxidative Stress-Induced Damage in Kidney Cells by Modulating Antioxidant, Longevity, and Senescence-Related Genes

**DOI:** 10.3390/biomedicines13123016

**Published:** 2025-12-09

**Authors:** Supansa Buakaew, Chadamas Sakonsinsiri, Worachart Lert-itthiporn, Ubon Cha’on, Tawut Rudtanatip, Ratthaphol Kraiklang, Waleeporn Kaewlert, Pornpattra Rattanaseth, Poungrat Pakdeechote, Raynoo Thanan

**Affiliations:** 1Department of Biochemistry, Faculty of Medicine, Khon Kaen University, Khon Kaen 40002, Thailand; supansabuakaew@kkumail.com (S.B.); schadamas@kku.ac.th (C.S.); woracle@kku.ac.th (W.L.-i.); ubocha@kku.ac.th (U.C.); walee@kkumail.com (W.K.); pornpattra_m@kkumail.com (P.R.); 2Chronic Kidney Disease Prevention in the Northeast of Thailand (CKDNET), Khon Kaen University, Khon Kaen 40002, Thailand; ratthaphol@kku.ac.th; 3Department of Anatomy, Faculty of Medicine, Khon Kaen University, Khon Kaen 40002, Thailand; tawut@kku.ac.th; 4Faculty of Public Health, Khon Kaen University, Khon Kaen 40002, Thailand; 5Department of Physiology, Faculty of Medicine, Khon Kaen University, Khon Kaen 40002, Thailand; ppoung@kku.ac.th; 6Center for Translational Medicine, Faculty of Medicine, Khon Kaen University, Khon Kaen 40002, Thailand

**Keywords:** oxidative stress, cellular senescence, antiaging, hesperidin, chronic kidney disease

## Abstract

**Background:** Oxidative stress arises from an imbalance between excessive oxidant production and impaired antioxidant defense systems. This imbalance leads to biomolecular damage, contributing to aging and age-related diseases such as chronic kidney disease (CKD). Oxidative stress is a well-established risk factor for CKD and has been reported to accelerate disease progression. Hesperidin, a flavanone glycoside abundant in citrus fruits, exhibits antioxidant, anti-hypertensive, and anti-inflammatory properties and has been suggested to attenuate CKD progression. However, its potential role in reversing oxidative damage in kidney cells remains unclear. **Methods:** This study aimed to investigate whether hesperidin can reverse oxidative damage in human kidney proximal tubular epithelial (HK-2) cells. Oxidative stress was induced by exposing HK-2 cells to 500 μM hydrogen peroxide (H_2_O_2_) for 6 h, followed by treatment with 100 μM hesperidin for 24 h. **Results:** Our results showed that hesperidin significantly ameliorated H_2_O_2_-induced cytotoxicity. In the hesperidin post-treatment group (H_2_O_2_ + hesperidin), the expression of the antioxidant gene manganese superoxide dismutase (MnSOD) and the longevity-associated gene sirtuin 1 (SIRT1) was upregulated, while the expression of the senescence-associated gene β-galactosidase was downregulated compared to the H_2_O_2_-only treatment. **Conclusions:** These findings suggest that hesperidin promotes recovery from oxidative injury in kidney cells by enhancing antioxidant and longevity pathways and reducing cellular senescence. This may contribute to improved renal health and potentially slow CKD progression in patients suffering from oxidative stress-related kidney damage.

## 1. Introduction

Chronic kidney disease (CKD) imposes a significant economic burden on both healthcare systems and patients [1]. CKD is defined as a persistent decline in kidney function and/or kidney structural abnormalities lasting at least 3 months. It is classified into three categories based on albuminuria and five stages according to glomerular filtration rate (GFR) [2]. GFR reflects the kidney’s filtration capacity, and its decline indicates nephron loss [3]. CKD symptoms vary by stage. In early stages (stages 1–3) with GFR ranging from 30–90 mL/min/1.73 m^2^, patients are often asymptomatic. In contrast, advanced stages (4–5; GFR < 30 mL/min/1.73 m^2^) may present with fatigue, anorexia, weight loss, pruritus, nausea, vomiting, muscle cramps, edema, and shortness of breath [2]. Modifiable CKD risk factors include diabetes, hypertension, and obesity; proper management of these factors can help slow disease progression [4]. To prevent CKD progression, patients are advised to limit sodium intake, control blood pressure and glucose levels, engage in regular physical activity, and follow a balanced diet [5]. In end-stage CKD, treatment options include dialysis or kidney transplantation; however, these approaches carry serious risks and complications [6].

Oxidative stress is characterized by an imbalance between pro-oxidant production and antioxidant defense systems, leading to the overproduction of reactive oxygen species (ROS) and reactive nitrogen species (RNS). It is known to increase with age and contributes to age-related diseases such as cancer and neurodegenerative disorders [7]. Oxidative stress plays a key role in cellular damage by inducing biomolecular oxidation, mitochondrial dysfunction, cell cycle arrest, and metabolic alterations, ultimately driving aging and disease progression [8]. CKD risk factors such as hypertension [9], obesity [10,11], aging [12], and diabetes [13] are also known to induce oxidative stress. Elevated oxidative stress is a critical contributor to CKD pathophysiology and is often observed in advanced stages of the disease [14]. These findings support the hypothesis that oxidative stress contributes to CKD progression and renal cell dysfunction via oxidative damage accumulation. Therefore, bioactive compounds capable of reversing oxidative damage in kidney cells could serve as promising therapeutic agents, particularly for early-stage CKD, to prevent further progression.

Hesperidin is a citrus bioflavonoid predominantly found in the peels and pulp of citrus fruits such as oranges, lemons, and bergamots, and it is abundant in members of the *Rutaceae* family. It is a flavanone glycoside composed of an aglycone moiety, hesperetin, and a glycoside, rutinoside [15]. It has been widely studied for its diverse pharmacological properties, including anti-inflammatory, antioxidant, anti-cancer, and anti-obesity activities, as well as its ability to lower cholesterol levels and blood pressure [16]. Hesperidin has shown the ability to scavenge ROS, modulate oxidative stress-related signaling pathways, and enhance endogenous antioxidant defenses in bovine mammary epithelial cells [17]. Growing clinical evidence supports the notion that hesperidin exerts protective effects in diabetes [18]. Furthermore, it has been reported to exert renoprotective effects by reducing oxidative damage and improving kidney function in cisplatin-induced kidney injury models [19]. However, the potential of hesperidin to reverse established oxidative damage in renal tubular epithelial cells remains underexplored. Understanding the molecular mechanisms underlying its ability to reverse oxidative damage may provide a rationale for its use as a therapeutic or preventive agent in oxidative stress-related renal disorders such as CKD.

This study aims to investigate whether hesperidin can reverse oxidative damage in kidney cells by activating genes associated with longevity and antioxidant defense while suppressing aging-related gene expression. Human kidney proximal tubular epithelial (HK-2) cells were selected as the model for this investigation. Proximal tubules are the primary site of kidney injury [20] and are highly vulnerable to oxidative stress due to their high metabolic demand and limited antioxidant capacity [21]. They are also highly susceptible to injury caused by urinary stone minerals, inflammatory cytokines, and oxidative stress, and such proximal tubular damage is strongly linked to lifestyle-related diseases, including diabetes and hypertension [22,23]. In addition, HK-2 cells have been widely reported to be sensitive to H_2_O_2_-induced oxidative injury and are therefore commonly used as an in vitro model for oxidative stress studies [24]. Therefore, HK-2 cells represent a suitable and physiologically relevant model for studying oxidative damage in the kidney.

To achieve this objective, an in vitro model of oxidative injury was established by exposing HK-2 cells to hydrogen peroxide (H_2_O_2_), followed by hesperidin treatment to assess its restorative effects. The expression of key genes involved in longevity and anti-aging (Klotho [KL] and sirtuin 1 [SIRT1]), antioxidant defense (MnSOD), and cellular senescence (β-galactosidase) was evaluated. MnSOD was selected as a major mitochondrial antioxidant enzyme responsible for detoxifying superoxide radicals [25]. SIRT1, a longevity-associated deacetylase, regulates oxidative stress responses, mitochondrial function, and cell survival [26]. Klotho, an anti-aging and renoprotective protein predominantly expressed in renal tissue, was included because of its central role in kidney injury and aging [27]. Senescence-associated β-galactosidase (SA-β-gal) was measured as a classical marker of oxidative damage–induced cellular senescence [28].

## 2. Materials and Methods

### 2.1. Cell Culture

HK-2 cells were cultured in Ham’s F-12 medium (Gibco/Life Technologies, Grand Island, NY, USA) supplemented with 10% fetal bovine serum, 100 µg/mL streptomycin, and 100 units/mL penicillin (complete medium). Cells were maintained at 37 °C in a humidified incubator containing 5% CO_2_ and 95% relative humidity.

### 2.2. H_2_O_2_ and Hesperidin Treatments

HK-2 cells were seeded into appropriate cell culture plates depending on the intended assay: 6-well plates at a density of 9 × 10^4^ cells/well for RNA and protein extractions, 48-well plates at a density of 7.5 × 10^3^ cells/well for immunocytochemistry, and 96-well plates at a density of 3 × 10^3^ cells/well for the SRB assay. Cells were incubated overnight in complete medium to allow for attachment. The following day, cells were treated with various concentrations of hydrogen peroxide (H_2_O_2_; Merck KGaA, Darmstadt, Germany) for 6 h to induce oxidative stress. After treatment, the medium was replaced with complete medium supplemented with various concentrations of hesperidin (Sigma-Aldrich, St. Louis, MO, USA), followed by incubation for an additional 24 h. The vehicle control group received complete medium containing 0.5% (*v*/*v*) DMSO. After the 24 h incubation period, cells were harvested for downstream analyses.

### 2.3. Sulforhodamine B (SRB) Assay

Following treatments, cells were fixed with 100 µL of 10% (*w*/*v*) trichloroacetic acid (TCA; AppliChem, Darmstadt, Germany) per well and incubated at 4 °C for 1 h. Fixed cells were washed three times with deionized water, air-dried, and incubated at 60 °C for 30 min. Cells were stained with 50 µL of 0.4% (*w*/*v*) SRB in 1% acetic acid (Sigma-Aldrich) for 45 min at room temperature in the dark. Excess dye was removed by washing three times with 1% acetic acid. Plates were dried again at 60 °C for 30 min, and bound dye was solubilized with 200 µL of 10 mM Tris base (pH 10.5). Plates were gently shaken for 1 h at room temperature, and absorbance was measured at 540 nm using a microplate reader (Sunrise™, Tecan, Zurich, Switzerland).

For data analysis, absorbance values (OD_540_) from all samples and blanks (medium-only wells) were collected. Blank values were averaged and subtracted from all sample readings to obtain net OD_540_. The mean ± SD was then calculated for each treatment group. Cell density was normalized to the mean OD_540_ value of the untreated control group (set to 1), and fold values were subsequently multiplied by 100 to express the results as percent of control.

### 2.4. Real-Time PCR

Total RNA was extracted using TRIzol™ reagent (Invitrogen, Carlsbad, CA, USA). Briefly, 1 mL of TRIzol™ was added to the cell pellet, and the lysate was homogenized by pipetting. The mixture was incubated at room temperature for 5 min, followed by the addition of 200 µL of chloroform. Samples were mixed gently, incubated for another 3 min, and centrifuged at 13,000 rpm for 20 min at 4 °C. The aqueous phase was transferred to a new 1.5-mL microcentrifuge tube, and 300–500 µL of cold isopropanol was added to precipitate RNA. After gentle mixing and incubation at room temperature for 20 min, samples were centrifuged at 13,000 rpm for 20 min at 4 °C. The resulting RNA pellet was washed with 1 mL of 70% ethanol and centrifuged at 13,000 rpm for 5 min at 4 °C. The supernatant was removed, and the pellet was air-dried at room temperature before being dissolved in 20–50 µL of DEPC-treated water and incubated at 4 °C overnight.

RNA concentration and purity were determined using a NanoDrop ND-2000 spectrophotometer (NanoDrop Technologies, Wilmington, DE, USA). Complementary DNA (cDNA) was synthesized from total RNA using the High-Capacity cDNA Reverse Transcription Kit (Applied Biosystems, Foster City, CA, USA) according to the manufacturer’s instructions. Quantitative real-time PCR (qRT-PCR) was performed using TaqMan™ gene expression assays (KL: Hs00183100_m1; SIRT1: Hs01009005_m1; MnSOD: Hs01553554_m1; GAPDH: Hs02786624_g1; Applied Biosystems, Foster City, CA, USA) on a QuantStudio 6 Flex Real-Time PCR System (Life Technologies, Tuas, Singapore). GAPDH served as the endogenous reference gene. Relative gene expression levels were calculated using the comparative Ct (2^–ΔΔCt^) method. The resulting values were expressed as fold change relative to the untreated control group, and these fold-change values were used for subsequent statistical analysis.

### 2.5. Western Blot Analysis

Cell pellets were lysed in radioimmunoprecipitation assay (RIPA) buffer containing a protease inhibitor cocktail. The lysates were thoroughly mixed and centrifuged at 15,000 rpm at 4 °C for 15 min. The supernatants were collected into new microcentrifuge tubes. Protein concentration was determined using the BCA protein assay kit (Bio-Rad Laboratories, Inc., Hercules, CA, USA), following the manufacturer’s instructions. An equal amount of protein from each sample was mixed with SDS loading buffer and boiled at 95 °C for 5 min. Proteins were separated by SDS-polyacrylamide gel electrophoresis (SDS-PAGE) and subsequently transferred onto polyvinylidene fluoride (PVDF) membranes (Merck KGaA, Darmstadt, Germany). The membranes were blocked in 5% (*w*/*v*) skim milk prepared in Tris-buffered saline containing 0.1% (*v*/*v*) Tween-20 (TBS-T) for 1 h at room temperature on a shaker. After blocking, the membranes were incubated overnight at 4 °C with specific primary antibodies against KL (1:2000; Cat. No. 28100-1-AP, Proteintech Group Inc., Rosemont, IL, USA), SIRT1 (1:2000; Cat. No. 13161-1-AP, Proteintech Group Inc., Rosemont, IL, USA), MnSOD (1:2000; Cat. No. 06-984, Merck KGaA, Darmstadt, Germany), and β-galactosidase (1:2000; Cat. No. 15518-1-AP, Proteintech Group Inc.), with gentle shaking. After washing, membranes were incubated with a horseradish peroxidase (HRP)-conjugated secondary antibody for 1 h at room temperature. Protein bands were detected using ECL™ Prime Western Blotting Detection Reagent (GE Healthcare UK Ltd., Little Chalfont, UK). Chemiluminescent signals were visualized and quantified using the Amersham ImageQuant 800 imaging system (Cytiva Life Sciences™, Marlborough, MA, USA). β-actin was used as an internal loading control.

Integrated density of Western blot bands was analyzed using ImageJ 1.53k software (National Institutes of Health, Bethesda, MD, USA). The density of each protein band was quantified and normalized to its corresponding β-actin band. The normalized values were then divided by the mean value of the untreated control group, which was set to 1, to obtain the relative fold change for each condition. These fold-change values were used for subsequent statistical analysis. The uncropped Western blot images are provided in Appendix A. Densitometric analysis of Western blot bands is shown in Appendix A.

### 2.6. Immunocytochemistry (ICC)

HK-2 cells (7.5 × 10^3^ cells/well) were seeded in 48-well plates and incubated overnight to allow for cell attachment. The following day, cells were treated with 500 μM hydrogen peroxide (H_2_O_2_) for 6 h to induce oxidative stress. After treatment, the medium was replaced with complete Ham’s F-12 medium containing 100 μM hesperidin and 0.5% (*v*/*v*) DMSO, and cells were further incubated for 24 h. After incubation, the cells were fixed with 4% (*w*/*v*) paraformaldehyde in phosphate-buffered saline (PBS) for 30 min at room temperature. The fixed cells were washed with PBS and permeabilized with 0.2% (*v*/*v*) Triton X-100 in PBS for 2 min at room temperature. After another PBS wash, cells were incubated with 0.3% (*v*/*v*) hydrogen peroxide (H_2_O_2_) in PBS for 30 min at room temperature to block endogenous peroxidase activity. To reduce non-specific antibody binding, cells were blocked with 3% (*w*/*v*) bovine serum albumin (BSA) in PBS for 30 min at room temperature. Cells were then incubated overnight at 4 °C with a rabbit polyclonal anti-β-galactosidase primary antibody (1:100 dilution, Cat. No. 15518-1-AP, Proteintech Group Inc.). After washing, cells were incubated with a horseradish peroxidase (HRP)-conjugated secondary antibody, and color development was performed using a DAB substrate kit (Vector Laboratories, Inc., Burlingame, CA, USA) according to the manufacturer’s instructions. Finally, the cells were dehydrated and observed under an inverted light microscope.

### 2.7. Statistical Analysis

All assays were performed with at least two independent experiments as noted in the figure legends. Statistical analyses were carried out using Microsoft^®^ Excel version 16.56 (Microsoft Corporation, Redmond, WA, USA). An independent Student’s *t*-test was used to compare differences between groups, and a *p*-value < 0.05 was considered statistically significant.

## 3. Results

### 3.1. Hesperidin Restored the Cell Density of Oxidatively Damaged Kidney Cells

The sulforhodamine B (SRB) assay was performed to evaluate the effects of hesperidin on HK-2 cell viability across a range of concentrations. HK-2 cells were treated with 0, 75, 100, 125, 150, 175, 200, 225, and 250 μM hesperidin and incubated for 24 h. The results showed that hesperidin exhibited no significant cytotoxicity at concentrations ranging from 75 to 250 μM. Moreover, a significant increase in cell density was observed at 100, 200, 225, and 250 μM compared to the untreated control (Figure 1A). Based on these findings and supporting evidence from the literature [17], 100 μM hesperidin was selected for subsequent experiments.

To determine the optimal concentration of the oxidative stress inducer, HK-2 cells were treated with varying concentrations of H_2_O_2_ (0, 100, 250, 500, 600, 700, 800, 900, and 1000 μM) for 6 h, followed either by immediate SRB assay or by replacing the medium with fresh complete media and further incubation for 24 h before performing the assay. The results demonstrated that H_2_O_2_ treatment for 6 h reduced cell density at concentrations between 500–1000 μM. Notably, a more substantial reduction in cell viability was observed in cells incubated for an additional 24 h post-H_2_O_2_ exposure, indicating that H_2_O_2_ exerts delayed cytotoxic effects during the recovery phase (Figure 1B). Based on these observations, 500 μM of H_2_O_2_ was selected as the optimal concentration for establishing an oxidative stress-induced cellular damage model in HK-2 cells.

To investigate the potential restorative effects of hesperidin, HK-2 cells were exposed to 500 μM H_2_O_2_ for 6 h to induce oxidative stress, followed by treatment with 100 μM hesperidin for 24 h (Figure 1C). As shown in Figure 1D, cell density was significantly reduced in the H_2_O_2_-only group compared to the untreated control. However, cells that received hesperidin following H_2_O_2_ exposure (H_2_O_2_ + Hes group) exhibited a significant increase in cell density compared to the H_2_O_2_-only treatment group, indicating a partial recovery. Despite this improvement, the cell density in the H_2_O_2_ + Hes group remained lower than that of the untreated control, suggesting that hesperidin could moderately restore kidney cells from oxidative damage.

### 3.2. Hesperidin Upregulated Longevity-Related Genes and Antioxidant Genes in Oxidatively Damaged Kidney Cells

The expression of KL mRNA was measured by real-time PCR. KL transcript levels were significantly upregulated in the hesperidin-treated group compared to the untreated control. In contrast, KL mRNA was significantly downregulated in the H_2_O_2_-only treatment group relative to the control. Post-treatment with hesperidin significantly increased KL mRNA expression compared to H_2_O_2_ treatment alone, indicating partial restoration of gene expression (Figure 2A). Western blot analysis further confirmed that KL protein levels were significantly increased in the hesperidin-treated group compared to the control (Figure 2B). However, under H_2_O_2_-induced damage conditions, hesperidin did not significantly improve KL protein levels, suggesting that oxidative stress may impair post-transcriptional regulation or protein stability despite mRNA recovery.

Regarding SIRT1, mRNA expression was significantly upregulated in the hesperidin post-treatment group compared to the H_2_O_2_-only group (Figure 2C). At the protein level, SIRT1 expression was elevated in hesperidin and H_2_O_2_ treatments compared to control cells, possibly as a compensatory mechanism. Post-treatment with hesperidin further enhanced SIRT1 protein expression, with a trend toward increased expression relative to the H_2_O_2_ group (Figure 2D). These findings suggest that hesperidin supports the recovery of oxidatively stressed HK-2 cells by promoting SIRT1 expression at both transcriptional and translational levels.

The MnSOD mRNA level was markedly elevated in the H_2_O_2_-exposed cells compared to the control cells, indicating activation of antioxidant defense pathways. Post-treatment with hesperidin significantly enhanced MnSOD mRNA expression even further compared to cells exposed to H_2_O_2_ alone (Figure 2E). Consistently, MnSOD protein expression was significantly upregulated in the hesperidin post-treatment group compared to the control group (Figure 2F). These results suggest that hesperidin enhances antioxidant defenses in oxidatively damaged kidney cells, potentially contributing to cellular recovery.

### 3.3. Hesperidin Reduced β-Galactosidase Expression in Kidney Cells Induced by Oxidative Stress

Western blotting and immunocytochemistry were employed to assess the expression of β-galactosidase, a senescence-associated marker, in HK-2 cells under different treatment conditions. The results showed that β-galactosidase protein expression was significantly upregulated in the H_2_O_2_-only group compared to the untreated control, indicating the induction of cellular senescence following oxidative stress. Post-treatment with hesperidin markedly reduced β-galactosidase expression compared to H_2_O_2_ treatment alone, as evidenced by both Western blotting and immunocytochemical staining (Figure 3A,B). These findings suggest that hesperidin exerts anti-senescent effects and promotes the recovery of oxidatively damaged kidney cells, potentially through the downregulation of β-galactosidase expression.

## 4. Discussion

Oxidative stress is a condition characterized by overproduction of oxidants, such as ROS, or a defect in the antioxidant system [29]. Oxidative damage resulting from this imbalance has been closely associated with the progression of CKD [30]. H_2_O_2_ is one type of ROS that generates hydroxyl radical (•OH) via Fenton’s reaction [31]. The •OH radical is the most reactive free radical that attacks a wide range of biomolecules [32]. A study on the cytotoxicity of H_2_O_2_ in human hepatocytes reported a reduction in cell viability following exposure to H_2_O_2_ [31]. Hesperidin has been reported to increase cell density in H_2_O_2_-pretreated hepatocytes [31]. Moreover, it possesses potent antioxidant properties and has been reported to decrease intracellular ROS levels induced by H_2_O_2_ treatment [17]. Consistent with these findings, the present results show a reduction in cell density following H_2_O_2_ exposure and its improvement by hesperidin, highlighting hesperidin’s ability to promote cellular recovery after oxidative damage. These observations prompted us to further examine key molecular pathways involved in stress recovery, particularly the longevity-associated genes KL and SIRT1, the senescence marker β-galactosidase, and the mitochondrial antioxidant enzyme MnSOD in oxidatively injured HK-2 cells.

Klotho (KL), a single-pass transmembrane protein that regulates aging-related processes through key pathways including phosphate homeostasis, insulin signaling, and Wnt signaling, is one of the major renal protective factors [33]. Previous studies have shown that oxidative stress can downregulate KL expression. For instance, oxidative stress induced by D-galactose (D-gal) treatment was reported to reduce KL protein levels in renal proximal tubular cells [34]. In contrast, baicalin, a flavone glycoside, was reported to restore KL expression and reduce oxidative stress under diabetes-induced kidney injury in a rat model [35]. These findings indicate that KL is highly susceptible to oxidative suppression, and its reduction is a well-recognized molecular feature of renal stress and premature aging. The ability of hesperidin to enhance KL expression under basal conditions, together with its partial restoration of KL levels in oxidatively injured cells, suggests that hesperidin may support renal recovery through KL-mediated pathways. This response implies that hesperidin can influence KL expression at both the transcriptional and translational levels. The incomplete restoration of KL protein under oxidative conditions may reflect delayed translation, accelerated protein turnover, or post-translational modifications, suggesting that a longer recovery period or alternative treatment strategies may be required to fully reestablish KL protein levels in damaged kidney cells.

Sirtuin 1 (SIRT1) is a nicotinamide adenine dinucleotide (NAD^+^)-dependent deacetylase that plays a central role in regulating gene expression, metabolic homeostasis, oxidative stress responses, and cellular longevity [36]. Previous studies have shown that SIRT1 expression is upregulated under oxidative stress conditions, particularly in response to hydrogen peroxide (H_2_O_2_), where it interacts with JNK1 to mediate cytoprotective effects [37]. Consistent with this concept, our findings support the view that SIRT1 acts as a key component of the cellular stress-adaptation machinery. Moreover, the further elevation in SIRT1 mRNA and protein levels following hesperidin treatment suggests that hesperidin may potentiate SIRT1-dependent pathways, thereby enhance cellular resilience and promote recovery from oxidative damage. Among the longevity-associated genes examined, SIRT1 exhibited the most robust response to hesperidin, underscoring its potential role as a primary mediator of hesperidin’s restorative effects.

Manganese superoxide dismutase (MnSOD, also known as SOD2) is a mitochondrial metalloenzyme essential for detoxifying superoxide radicals and protecting cells from oxidative stress-induced injury [25]. Its expression is known to increase in response to mitochondrial ROS production, such as in lipopolysaccharide (LPS)-treated microglial cells [38]. Additionally, natural antioxidant compounds, including flavonoid-rich extracts from *Centella asiatica*, have been reported to enhance SOD2 expression in H_2_O_2_-stressed human fibroblasts [39]. In line with these observations, our results indicate that hesperidin amplifies the MnSOD response beyond the endogenous upregulation triggered by H_2_O_2_ alone. This enhancement suggests that hesperidin strengthens mitochondrial antioxidant defenses by promoting MnSOD expression, thereby improving the oxidative stress resistance of damaged kidney cells.

The coordinated regulation of KL, SIRT1, and MnSOD suggests an interconnected protective axis. SIRT1 activates FOXO3a to induce MnSOD expression, thereby enhancing mitochondrial ROS detoxification [40]. KL has also been shown to regulate SIRT1 expression, as KL deficiency leads to reduced SIRT1 levels in several models [41]. The simultaneous upregulation of KL, SIRT1, and MnSOD following hesperidin treatment suggests that hesperidin may promote oxidative stress recovery through a coordinated activation of longevity and antioxidant pathways in HK-2 cells.

β-Galactosidase is widely recognized as a biomarker of cellular senescence, and its expression is commonly elevated in response to oxidative stress [42]. Consistent with this concept, previous work has shown that luteolin, a naturally occurring flavonoid, can attenuate β-galactosidase activity in H_2_O_2_-treated mouse auditory cells, thereby exerting anti-senescent effects [43]. This highlights the sensitivity of β-galactosidase to oxidative injury and its usefulness as an indicator of stress-induced premature aging. In our model, oxidative stress triggered a senescence-like response in HK-2 cells, as reflected by increased β-galactosidase expression. The ability of hesperidin to reduce this elevation suggests that it suppresses stress-induced senescence, complementing its effects on KL, SIRT1, and MnSOD. When considered together, these coordinated responses indicate that hesperidin may enhance cellular resilience and mitigate premature aging by modulating key regulators of longevity and oxidative stress defense pathways.

This study has certain limitations. The in vitro HK-2 model cannot fully recapitulate the complexity of renal physiology, and therefore in vivo studies are warranted. However, hesperidin has already been evaluated in clinical trials showing improvements in obesity [44] and hypertension [45], both of which help inhibit the development and progression of CKD. It has also been reported to be non-toxic at doses up to 2000 mg/kg in rats [46,47,48], supporting its favorable safety profile for potential human use. Furthermore, orally administered hesperidin has been shown to ameliorate proximal tubular epithelial cell injury induced by advanced glycation end products [22]. Although the oral bioavailability of hesperidin is limited, drug-delivery strategies, such as lipid-based formulations, polymeric nanoparticles, and phytosomes, may enhance its absorption and therapeutic efficacy [49]. Moreover, combining hesperidin with other phytochemicals has demonstrated greater benefits in improving metabolic disorders [50] as well as lipid profile and BMI in healthy subjects [51]. Collectively, these findings suggest that hesperidin may serve as a promising therapeutic agent for preventing CKD progression.

## 5. Conclusions

In summary, hesperidin enhanced the viability of HK-2 cells following oxidative stress-induced injury. Its ability to reverse oxidative damage appears to be mediated through multiple molecular mechanisms, including the upregulation of KL and SIRT1, and the suppression of senescence-associated β-galactosidase expression. Additionally, hesperidin improved the antioxidant defense system by upregulating MnSOD expression, a key mitochondrial enzyme responsible for neutralizing ROS (Figure 4). These findings suggest that hesperidin, a naturally occurring plant-derived flavonoid, may partially reverse oxidative damage and modulate longevity and stress-response pathways in kidney cells. Although the degree of recovery observed was modest, this study provides a promising foundation for considering hesperidin as a potential adjunctive agent to delay CKD progression. Further investigations, including dose–response studies, long-term exposure analysis, and validation in animal models or clinical settings, are warranted to fully elucidate its therapeutic potential and optimize its application.

## Figures and Tables

**Figure 1 biomedicines-13-03016-f001:**
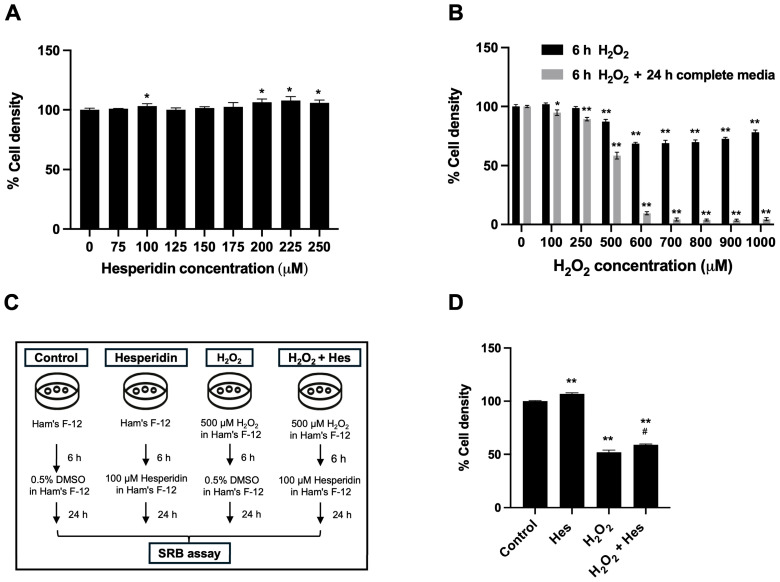
Effects of H_2_O_2_ and hesperidin on HK-2 cell density measured by SRB assay. All data are presented as mean ± SD of the percent cell density normalized to untreated control cells, calculated from OD_540_ values obtained from the SRB assay. (**A**) Cytotoxicity of hesperidin on HK-2 cells. Cells were treated with 0–250 µM hesperidin for 24 h (*n* = 3 per group), based on two independent experiments. (**B**) Cytotoxicity of hydrogen peroxide. HK-2 cells were treated with 0–1000 µM H_2_O_2_ for 6 h (black bars) or treated for 6 h followed by recovery in complete medium for 24 h (gray bars) (*n* = 3 per group), based on three independent experiments. (**C**) Schematic diagram of experimental treatment conditions: untreated control, 100 µM hesperidin (Hes), 500 µM H_2_O_2_, and co-treatment with H_2_O_2_ followed by hesperidin (H_2_O_2_ + Hes). (**D**) Comparison of cell density among treatment groups (*n* = 3 per group), based on three independent experiments. Statistical analysis was performed using an independent Student’s *t*-test. * *p*-value ≤ 0.05 compared to control. ** *p*-value ≤ 0.001 compared to control. ^#^
*p*-value ≤ 0.05 compared to the H_2_O_2_-treated group.

**Figure 2 biomedicines-13-03016-f002:**
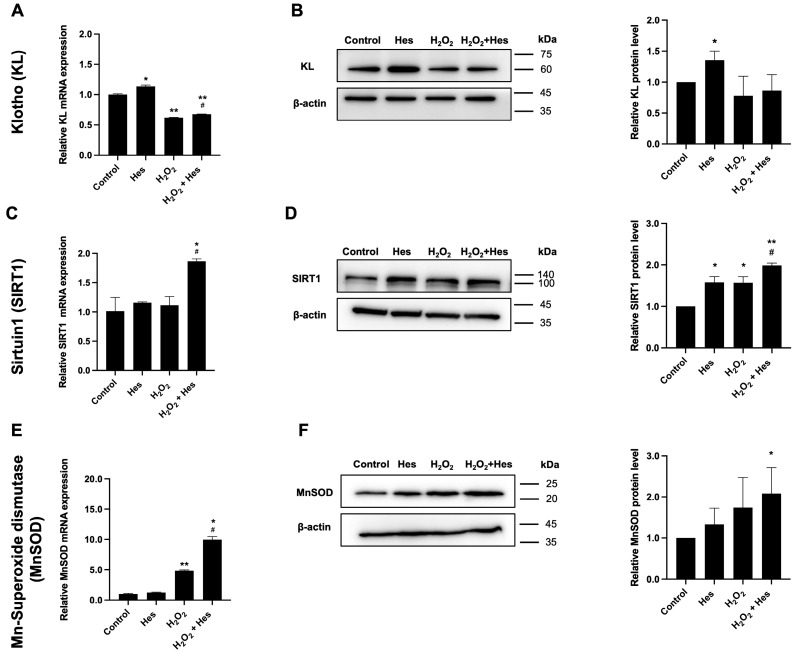
Effects of hesperidin on KL, SIRT1, and MnSOD mRNA and protein expression levels in oxidatively damaged HK-2 cells induced by H_2_O_2_. (**A**,**C**,**E**) Relative fold changes in KL, SIRT1, and MnSOD mRNA expression normalized to GAPDH, as measured by real-time PCR (*n* = 2 per group). (**B**,**D**,**F**) Relative fold changes in protein expression levels of KL (*n* = 3 per group), SIRT1 (*n* = 3 per group), and MnSOD (*n* = 4 per group) normalized to β-actin, as determined by Western blot analysis. Statistical analysis was performed using an independent Student’s *t*-test. kDa represents kilodalton. * *p*-value ≤ 0.05 compared to untreated control. ** *p*-value ≤ 0.001 compared to untreated control. ^#^ *p*-value ≤ 0.05 compared to H_2_O_2_-treated group.

**Figure 3 biomedicines-13-03016-f003:**
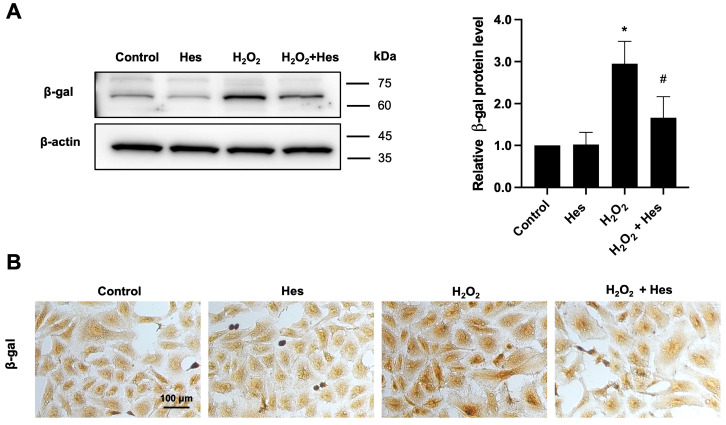
Effects of hesperidin on β-galactosidase protein expression levels detected by Western blot analysis and immunocytochemistry in oxidatively damaged HK-2 cells induced by H_2_O_2_. (**A**) Relative fold changes in protein expression level of β-galactosidase (β-gal) normalized by β-actin (*n* = 3 per group). kDa represents kilodalton. (**B**) Immunocytochemistry staining results of β-galactosidase. Statistical analysis was performed using an independent Student’s *t*-test. * *p*-value ≤ 0.05 compared to control. ^#^ *p*-value ≤ 0.05 compared to the H_2_O_2_-treated group.

**Figure 4 biomedicines-13-03016-f004:**
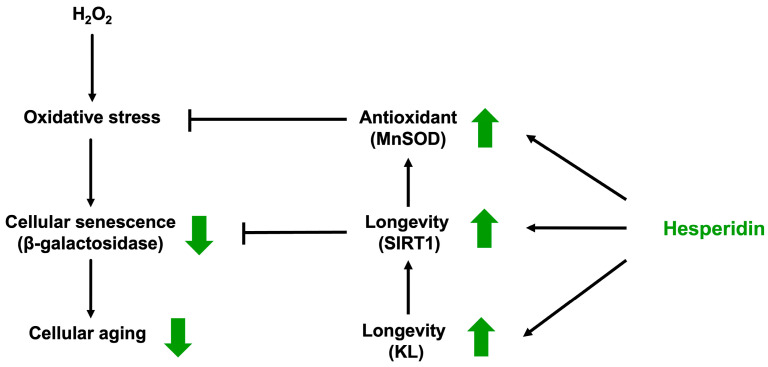
Proposed effects of hesperidin on the recovery of kidney cells from oxidative damage. The green arrows indicate the stimulatory (↑) or inhibitory (↓) effects mediated by hesperidin on the indicated pathways.

## Data Availability

All data are included in this article or Appendix A.

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
