# Peer review of "Hesperidin Reverses Oxidative Stress-Induced Damage in Kidney Cells by Modulating Antioxidant, Longevity, and Senescence-Related Genes"

_biomedicines, 2025, doi:10.3390/biomedicines13123016_

Round 1

Reviewer 1 Report

Comments and Suggestions for Authors

This manuscript addresses a significant and clinically relevant issue: the role of oxidative stress in the pathogenesis of kidney disease. The work is original, well-designed, and based on solid methodology. Nevertheless, some aspects require revision to improve clarity, scientific rigor, and presentation.

The study presents an interesting and relevant objective, focusing on the analysis of the effect of hesperidin on gene expression related to antioxidant defence and longevity in a model of kidney damage. The choice of this flavonoid, as well as the molecular approach adopted, are appropriate.

The figures are clear, informative, and well organised. They allow the logic of the results presented to be followed adequately.

It was a good decision to evaluate protein expression using two different techniques. This approach adds robustness and credibility to the results obtained.

Even so, some considerations may be taken into account to improve overall understanding of the work.

  1. It would be advisable to justify more clearly why the study focuses specifically on kidney proximal tubular epithelial cells. This methodological decision may have relevant implications for the interpretation and applicability of the results.
  2. It is mentioned that 9×10⁴ cells were seeded per well. An explanation of this specific cell concentration would be appreciated. Is it based on previous studies, a growth curve, or the authors' own experimental experience?
  3. It is important to clearly specify the statistical analyses used in the study in statistics section, including which tests were used.
  4. The use of the term ‘H₂O₂-treated’ in the results body may be confusing, as it can be misinterpreted as a combined treatment with hesperidin. It would be advisable to rephrase it for clarity.
  5. It would be appropriate to include a brief justification for the selection of the genes. This would help to reinforce the connection between the study objectives and the molecular markers chosen.

The inclusion of Klotho as a marker is viewed positively, given its key role in renal physiology and ageing processes. Its selection demonstrates an intelligent and well-founded approach.

  1. It would be useful to mention whether hesperidin has been used previously in clinical or preclinical studies in humans, and whether there is evidence of its safety and efficacy in related conditions.
  2. Finally, it would be advisable to include a brief section on the limitations of the study.

Reviewer 2 Report

Comments and Suggestions for Authors

The work is devoted to a study that complements the research of other authors on the antioxidant effect of hesperidin. It can be used as the first rough experiments for further work on the possible full search for evidence of the possibility of using hesperidin in practical healthcare for the treatment of some kidney diseases that occur against or for the reasons of the development of oxidative stress.

It would be useful to supplement the article with the following additions:

  1. In the captions to the figures, it is necessary to specify:

1а- in what units the data are presented (such as average values and standard errors)

1b- How many independent measurements were used to calculate these values.

  1. In the Materials and Methods section 2.7. Statistical analysis, and in the captions of the figures, it is necessary to specify which method was used to evaluate the differences between the groups (t-test, ANOVA, etc.).
  2. Specify the limitations of the conclusions made in the work.

Reviewer 3 Report

Comments and Suggestions for Authors

The manuscript addresses an interesting and relevant topic concerning the protective effects of hesperidin on oxidative stress-induced damage in renal tubular epithelial cells. The study is well-motivated, with a clear rationale linking oxidative stress, cellular senescence, and kidney pathophysiology. The introduction provides a strong contextual framework supported by up-to-date references, successfully establishing the significance of exploring natural antioxidants for chronic kidney disease prevention and management.

The experimental design is generally appropriate, using the HK-2 cell model to simulate oxidative injury and test the recovery potential of hesperidin. However, the Materials and Methods section could benefit from minor improvements in clarity and reproducibility. For instance, the number of biological replicates, details on normalisation procedures, and statistical tests used should be stated more explicitly. The statistical analysis is described very briefly; specifying whether the p-values were derived from t-tests or ANOVA would improve transparency.

The results are clearly presented and logically structured, with adequate figure quality and supporting legends. The integration of mRNA, protein, and immunocytochemistry data provides good internal consistency. However, the authors might consider including representative Western blot images with molecular weight markers and quantitative data in supplementary materials to enhance the rigour of the findings. It would also be helpful to clarify whether the observed changes in KL and SIRT1 expression were normalised to untreated controls or to the oxidative stress group.

The discussion is comprehensive and well-linked to the presented data. The authors successfully relate their results to previous findings, emphasising hesperidin’s antioxidant and anti-senescent roles. I suggest that the discussion could be slightly condensed by avoiding repetition of the results, and the authors might elaborate briefly on the mechanistic interplay between SIRT1, KL, and MnSOD pathways in oxidative stress recovery, which would strengthen the mechanistic insight of the work.

Comments on the Quality of English Language

The English language is generally understandable but could be improved to enhance clarity and fluency. Minor grammatical corrections, sentence restructuring, and consistency in tense (especially between Results and Discussion) would improve readability. Some sentences are overly long or repetitive, which slightly affects the flow.

Round 2

Reviewer 2 Report

Comments and Suggestions for Authors

The manuscript has been substantially improved. The authors have addressed all my comments, and their manuscript can be recommended for publication.